# Lectin Sequence Distribution in QTLs from Rice (*Oryza sativa*) Suggest a Role in Morphological Traits and Stress Responses

**DOI:** 10.3390/ijms20020437

**Published:** 2019-01-20

**Authors:** Mariya Tsaneva, Kristof De Schutter, Bruno Verstraeten, Els J.M. Van Damme

**Affiliations:** 1Laboratory of Biochemistry and Glycobiology, Department of Biotechnology, Faculty of Bioscience Engineering, Ghent University, Coupure Links 653, 9000 Ghent, Belgium; Mariya.Tsaneva@UGent.be; 2Laboratory of Agrozoology, Department of Plants and Crops, Faculty of Bioscience Engineering, Ghent University, Coupure Links 653, 9000 Ghent, Belgium; Kristof.DeSchutter@UGent.be; 3Laboratory of Epigenetics and Defense, Department of Biotechnology, Faculty of Bioscience Engineering, Ghent University, Coupure Links 653, 9000 Ghent, Belgium; Bruno.Verstraeten@UGent.be

**Keywords:** lectin, QTL, resistance/tolerance, morphology, phylogeny, protein domain

## Abstract

Rice (*Oryza sativa*) is one of the main staple crops worldwide but suffers from important yield losses due to different abiotic and biotic stresses. Analysis of quantitative trait loci (QTL) is a classical genetic method which enables the creation of more resistant cultivars but does not yield information on the genes directly involved or responsible for the desired traits. Lectins are known as proteins with diverse functions in plants. Some of them are abundant proteins in seeds and are considered as storage/defense proteins while other lectins are known as stress-inducible proteins, implicated in stress perception and signal transduction as part of plant innate immunity. We investigated the distribution of lectin sequences in different QTL related to stress tolerance/resistance, morphology, and physiology through mapping of the lectin sequences and QTL regions on the chromosomes and subsequent statistical analysis. Furthermore, the domain structure and evolutionary relationships of the lectins in *O. sativa* spp. indica and japonica were investigated. Our results revealed that lectin sequences are statistically overrepresented in QTLs for (a)biotic resistance/tolerance as well as in QTLs related to economically important traits such as eating quality and sterility. These findings contribute to the characterization of the QTL sequences and can provide valuable information to the breeders.

## 1. Introduction

Rice (*Oryza sativa*) is one of the major cereal crops worldwide and is the main food source for more than half of the human population around the world [1,2]. It is predicted that by 2050 the general agricultural production should increase by 60% to cover the food requirements [3]. In particular, the rice demand of Asia is expected to be 70% higher within 30 years [4]. At the same time, 50% to 90% of the rice production worldwide is annually destroyed or diminished due to drought, salt stress, rice blast disease, etc., or a combination of these major stresses. Ongoing research aims to define the genes/proteins that contribute to the resistance or tolerance of rice towards a broad spectrum of abiotic and biotic stresses. The combined use of multiple genes conferring tolerance/resistance to one or a couple of stress factors (so-called pyramiding of genes) is considered a promising approach for the elaboration of resistant rice cultivars with high yields but at the same time is a very challenging task for the breeders [5,6].

With more than 120,000 varieties [7], rice is a cereal which grows in diverse environmental conditions with respect to water supply, climate, etc. [3]. Rice cultivars are mainly grouped into two subspecies, indica and japonica, which most likely originate from different ancestors and have been domesticated independently in India and China, respectively [1]. The large diversity in the growth conditions highly affects and assists the divergence between these two subspecies, leading to differences in morphological characteristics as well as in seed quality and stress resistance [1,2,8], with a higher diversity observed in the japonica subspecies [7]. In addition, several reports uncovered differences at the transcriptome, proteome, and metabolome level of both subspecies [1,2].

The choice of rice cultivar grown by the farmer is usually based on yield and grain quality. Unfortunately, these high-yielding cultivars can be more susceptible to diseases while other cultivars combine resistance towards a couple of abiotic and biotic stresses but do not possess the desired seed quality [3]. Stress tolerance, pathogen resistance and grain yield and quality are complex traits which involve multiple genes located in different quantitative trait loci (QTLs). QTL analysis is complicated by the fact that one QTL region can have pleiotropic effects on a wide range of traits, or consists of multiple QTLs. In addition, QTLs are frequently subjected to epistatic interactions and their traits can be significantly influenced by the environment [3,9].

Multiple research groups focus their work and efforts on the investigation of QTLs related with different (a)biotic stresses but only a few publications address the combination of multiple stresses and search for combined tolerance/resistance [3]. A large part of the QTL research data is gathered in two databases which offer a QTL repository: Gramene (http://archive.gramene.org/qtl/) and Q-TARO (QTL Annotation Rice Online, http://qtaro.abr.affrc.go.jp/) database. While Gramene comprises QTL information for eight monocots and also integrates the data from Q-TARO [10]; Q-TARO is a specialized rice database which has collected more than 1,000 QTLs. After analysis, the QTLs are grouped into four major categories, namely morphological traits, physiological traits, resistance or tolerance, and others [11].

Even though QTL analysis has been used widely in plant breeding for decades, the exact genes that confer or contribute to a particular trait are rarely known, cloned, or characterized, especially in complex traits such as drought tolerance [12]. Nevertheless, for genes related to grain number, submergency or salinity tolerance, lodging resistance, and disease resistance there are some examples where particular genes have been associated with specific traits [9]. Among these are two QTLs where lectin genes have been shown to play an important role. The Jacalin-related lectin SaLT was reported to be part of SalTol, a QTL important for salt stress tolerance [13]. The *Galanthus nivalis* agglutinin (GNA)-related lectin *Pi-d2* is one of the genes important for resistance against rice blast [14]. Additionally, lectin overrepresentation was reported in QTLs for nematode resistance [15].

Lectins are carbohydrate-binding proteins, they bind specifically and reversibly to sugars or more complex carbohydrates but their carbohydrate binding domain does not possess any enzymatic activity. The classification of plant lectins in different families is based on sequence relationships, rather than on carbohydrate-binding properties. Indeed, all lectin motifs are characterized by specific sequences, which after three-dimensional folding will yield a lectin or carbohydrate recognition domain with one or more carbohydrate-binding sites. In principle, each lectin family groups all proteins possessing one particular lectin domain based on sequence similarity. Different lectin motifs differ from each other in their sequences but can show reactivity towards similar carbohydrate structures, indicating that specificity is not linked to the occurrence of one particular lectin motif. Furthermore, the binding site for multiple lectin motifs was shown to be extended, enabling not only the binding of monosaccharides but allowing anchoring of elongated carbohydrate structures. Each lectin family contains at least one or a few lectins that have been studied in detail for their carbohydrate-binding properties. These data show that the specificity of some lectin domains is rather strict whereas other lectin domains can interact with a diverse range of carbohydrate structures [16].

The large majority of the lectins studied consists of one or more lectin domains [16]. Genome studies revealed that lectin motifs are frequently linked with other protein domains, such as protein kinases, glycosyl hydrolases, dirigent or F-box domains, resulting in multi-domain proteins [17]. Although several reports suggest an important role for lectins in plant defense and innate immunity [17,18,19,20], the physiological importance of most lectins remains poorly characterized. Based on their subcellular localization and expression levels, lectins can be divided into two large groups. The first group contains abundant lectins that localize to the vacuole or to the extracellular compartment whereas the second group encompasses lectins located in the cytosolic or nuclear compartment that are expressed mainly after exposure of the plant to a particular stress stimulus. Based on the sequence similarity between the lectin domains, the whole group of plant lectins can be organized in twelve lectin families, in particular *Agaricus bisporus* agglutinin homologs, Amaranthins, Class V chitinase homologs with lectin activity (CRA), Cyanovirin family, *Euonymus europaeus* agglutinin family (EUL-related lectins), *Galanthus nivalis* agglutinin (GNA) family, Heveins, Jacalin-related lectins, Legume lectins, LysM domain lectin family, Nictaba family, and RicinB family [16].

Several reports investigated the distribution and phylogeny of plant lectin motifs in the model species *Arabidopsis thaliana* [21], the medicinal plant *Morus notabilis* (mulberry) [22], and crops such as *Glycine max* (soybean) and *Cucumis sativus* (cucumber) [23,24] or performed a comparative analysis between several species, also including rice [20,25]. All reports provide proof from literature and in silico data for the involvement of lectins in sensing or responding to different (a)biotic stresses. Nevertheless, the function of most lectin genes remains unclear. 

In this project, the occurrence of lectin genes in rice QTLs related to resistance/tolerance against biotic and abiotic stress as well as in QTLs associated with morphological and physiological traits was investigated. Furthermore, the genomic and phylogenetic analysis as well as the domain structure for different lectins in two rice subspecies are reported. This study contributes to the characterization of QTL regions and generates knowledge on the physiological role of lectins in plants and their importance for plant growth, development, and defense.

## 2. Results

### 2.1. Lectin and QTL Distribution in Rice Genomes

The rice genome has been sequenced from two independent sequencing projects, in particular, the MSU Rice Genome Annotation Project (Michigan State University, http://rice.plantbiology.msu.edu/) and the Rice Annotation Project Database (RAP-DB) (International Rice Genome Sequencing Project, https://rapdb.dna.affrc.go.jp/http://rice.plantbiology.msu.edu/). A previous search for lectin families in rice using the MSU annotation database identified nine lectin families in *O. sativa* spp. japonica including homologs of the class V chitinases, *Euonymus*-related lectins, GNA-related lectins, Heveins, Jacalin-related lectins, Legume lectins, LysM domain lectins, Nictaba-like lectins, and RicinB lectins [20,26]. A comparative analysis between the MSU and RAP-DB annotation databases led to significant differences in the number of lectin sequences ranging from 329 according to MSU to 291 sequences in RAP-DB. This difference in the number of identified lectin motifs is mainly due to a reduction in the number of sequences encoding Nictaba-like lectins, GNA-related lectins, and Legume lectins, see Appendix A. Furthermore, none of the sequences containing RicinB lectin domains could be retrieved from RAP-DB.

To investigate the distribution of the lectin genes throughout the rice (*O. sativa* spp. japonica) genome sequences with lectin motifs were mapped onto the 12 rice chromosomes, see Figure 1. Although lectin sequences are distributed over all chromosomes, the lectin density for each chromosome is quite variable. Three chromosomes hold 45% of all the lectin sequences while representing just 29.06% of the genome size. Specifically, chromosomes 4, 1, and 7 hold 20.9%, 12.6%, and 11.7% of all the lectin sequences, respectively, while these chromosomes account for 9.51%, 11.59%, and 7.96% of the genome size. Chromosomes 5, 12, and 3 carry the lowest number of lectin sequences (4.3%, 5.2%, and 5.5%, respectively) while representing 25.16% of the genome size (8.03%, 7.38%, and 9.76%, respectively, see Appendix A. Most lectin genes occur in condensed blocks, with 62.8% of the lectin sequences being present in tandem repeats. In total, 54 tandem duplication blocks have been identified throughout the genome containing 125 lectin genes. In addition, 14 segmental duplications were mapped containing 33 lectin genes, see Appendix A.

A comparative analysis between the *O. sativa* subspecies japonica and indica revealed differences in the number of lectin sequences. Although for most lectin families the differences are minor, important differences in the number of lectin genes as well as in their domain organization were observed mainly for the GNA family, see Table 1 and Appendix A. In addition, some unique protein domain combinations were retrieved only for one of the subspecies, for example, in the GNA, Jacalin-related lectin, Legume lectin, LysM and Nictaba families of rice subspecies indica, see Appendix A. With the exception of CRA, all lectin families comprise not only single lectin domain sequences but also sequences where the lectin domains are tandemly repeated. The presence of two or three lectin domains is frequently observed in the families of the Jacalin, LysM, and EUL lectins and rarely in GNA, Legume lectin, and Hevein families. The combination of lectin motifs with a protein kinase domain is present in most rice lectin families, see Table 1. The F-box domain is also associated with more than one lectin family, in particular, Nictaba and LysM lectins. Equal numbers of lectin sequences of the Hevein and EUL lectins were retrieved for the subspecies japonica and indica, suggesting that these sequences are highly conserved in rice. Within the EUL family and the CRA family, all identified lectin sequences only contain lectin domains. The EUL lectins are composed of a single or two lectin domain sequences. It was reported before that tandem arrayed EUL domain sequences are typical for many monocotyledonous plants but not for dicotyledonous plants [26].

In this study, we performed an extensive analysis on the lectin distribution in the rice genome and their association with QTL regions to elucidate whether lectin sequences are widely and significantly present in QTL regions. Therefore, QTLs retrieved from the Q-TARO database were mapped onto the rice chromosomes, see Figure 1. A comparative analysis between the distribution of lectin genes and of QTLs on different chromosomes revealed that a large number of lectin sequences are located within QTL regions, especially on chromosomes 1, 4, and 6. These QTLs with lectin sequences belong to all four groups (morphological traits, physiological traits, tolerance/resistance and others). Lectin sequences from all lectin families (except the CRA family) are present in the four major categories of QTLs, see Appendix A.

### 2.2. Lectin Sequences Are Widely Distributed in QTLs for Resistance or Tolerance

The presence of lectin sequences in QTLs attributed to resistance/tolerance was analyzed in detail since lectins have been associated with plant defense and the perception of pathogens or abiotic stress in multiple publications [17,19,27]. Many rice lectin sequences are located in QTL regions annotated as being responsible for resistance/tolerance, the percentage of lectin sequences in QTLs related to abiotic stresses (ranging from 40% to 100% for different lectin families) being markedly higher than those attributed to biotic stresses (10–43% for different lectin families, see Appendix A).

#### 2.2.1. Abiotic Stress

All sequences containing QTLs known to be associated with abiotic stress factors (salinity tolerance and submergency tolerance) contain at least one lectin gene, and at least 50% of the QTLs for other abiotic stress factors contain at least one lectin gene, see Figure 2. GNA-related lectin genes are the only lectin family present in QTL regions attributed to lodging resistance, see Appendix A.

Salinity and drought are among the most devastating abiotic stresses for rice [5,6]; consequently, lectin distribution in QTLs for these traits was investigated in detail. Chromosome 1 contains multiple overlapping QTLs related to salinity whereas both chromosomes 1 and 3 include QTLs for drought, see Figure 1 [11]. All 39 lectin sequences on chromosome 1 are present in at least one QTL, and 37 of them are in QTLs for abiotic stress, in particular, 33 lectin sequences appeared in QTL regions for drought tolerance and in QTL regions associated with salinity tolerance, see Appendix A. Our analysis showed a significant overrepresentation of lectin sequences only in the traits drought tolerance and soil stress tolerance but not in the trait for salinity tolerance, see Figure 2 and Appendix A.

SaLT (or Orysata, Os01g0348900), a Jacalin-related lectin that confers salinity tolerance [13] is localized in a QTL for salt stress. Both SaLT and OsJAC1, another Jacalin-related lectin, appeared in QTLs for drought on chromosome 1 and 12, respectively, in our study. Multiple lectin sequences are present in more than one type of QTL, see Appendix A, the highest number of lectin sequences (35) appearing simultaneously in QTLs for drought and soil stress. 

#### 2.2.2. Biotic Stress

Among the QTLs related with biotic stress, sheath blight resistance showed the highest percentage of lectin sequences (80%), followed by blast resistance, insect resistance, and bacterial blight resistance, all of them having lectin genes in less than 50% of the QTLs for these traits, see Figure 2. From all these traits, only the QTLs for blast resistance showed a significant overrepresentation of lectin genes, see Appendix A. The most prevalent family in this trait is the GNA family with seven genes, followed by the Jacalins (five genes with four of them containing a dirigent domain, including OsJAC1) and LysM domains (four members). Although the presence of lectins in QTLs for sheath blight resistance, insect resistance, and bacterial blight resistance was not statistically significant, our analysis showed the presence of the lectins CeBiP and OsLYP4 (Os09g0452200) in QTLs for sheath blight resistance and OsCERK in the QTL for insect resistance and bacterial blight resistance. OsCERK1 is also localized in a QTL related to roots, which is the location of the symbiotic interaction. These data are consistent with the functional characterization of the gene. Our analysis showed that OsLYP4 appeared together with CeBiP in a QTL annotated as being important for the response to another fungal pathogen (*Rhizoctonia solani*) causing sheath blight disease.

Jacalin-related lectins containing a dirigent domain are considered as important players in plant innate immunity since the dirigent domain was reported to be involved in plant defense and experimental data show its involvement in the response to *Magnaporte grisea* [28]. OsJAC1 was identified in a QTL for rice blast disease together with the rest of the Jacalins containing a dirigent domain which is in agreement with the literature; however, SaLT was not identified in any QTL related to insect resistance, contrary to some reports [29]. Additionally, it has been reported that lectin genes are statistically significantly overrepresented in six out of 11 QTLs associated with resistance towards the root-knot nematode *Meloidogyne graminicola* [15], see Appendix A.

The appearance of the same lectins in QTLs for abiotic, as well as biotic stress, was observed a couple of times. For example, our analysis shows that 23 lectins which were represented in QTLs for cold tolerance also appeared in QTLs for insect resistance. In the case of bacterial blight disease this observation is even more striking because 12 of the 16 lectins present in these QTLs also appeared in QTLs for cold tolerance and 15 genes were found in QTLs for drought tolerance, see Appendix A. Partial co-appearance was observed also between lectins in QTLs for drought tolerance and sheath blight resistance (nine lectins) and for lectins in QTLs for soil stress tolerance and blast resistance (13 lectins).

### 2.3. Lectins Are Widely Distributed in QTLs for Morphological and Physiological Traits

The stress response and the adaptation of a plant are intimately linked with the morphological and physiological traits of the investigated plant species or even cultivar. A large number of lectin sequences has been identified in QTLs for all six categories related to morphological traits (culm/leaf, dwarf, panicle/flower, seeds, root and shoot/seedling), see Figure 3. Lectin sequences from all the lectin families, except for the two smallest families CRA and EUL, are present in QTLs for all six morphological traits, see Appendix A. For all traits, at least 50% of the QTLs contain lectin sequences, see Figure 3. Furthermore, lectin sequences are statistically overrepresented in the QTLs related to categories culm/leaf, panicle/flower, seeds, and shoot/seedling, see Appendix A.

A high percentage of lectin-containing QTLs (at least 50%) was observed in all categories of physiological traits, especially in the category lethality (83%). The presence of lectin sequences is statistically significant for the traits eating quality and sterility, see Figure 3. Lectin sequences of different lectin families are present in QTLs associated with morphological and physiological traits and their occurrence is proportional to the size of the lectin family (number of lectin sequences), see Appendix A.

The association between traits such as seeds and eating quality and panicle/flower and sterility is a good indication of the importance of lectins in plant reproduction and the marketing or seed quality and deserves further investigation at both a molecular and breeding level. An excellent example which illustrates these connections is a gene encoding a LysM domain protein called OsEMSA1 (Os10g0524300). In our analyses, OsEMSA1 appeared in QTL for panicle, seeds, and sterility but not in any other QTL for morphological/physiological traits or QTL for tolerance/resistance. A GNA gene called OsLSK1 (Os01g0669100) and its six homologs in the rice genome (Os01g0668400, Os01g0668600, Os01g0668901, Os01g0670100, Os01g0670300, and Os01g0670600) appeared in a QTL for seeds, and more specifically in a QTL responsible for grain yield, as well as in QTLs for drought tolerance and salinity tolerance, which is in agreement with the literature [30]. A search in RAP-DB identified two paralogs for OslecRK, a GNA containing lectin receptor-like kinase in subspecies indica which plays a role in seed germination and plant immunity [31], in *O. sativa* japonica (Os06g0620200 and Os08g0230800). Os06g0620200 appeared in QTLs for dwarfs, roots, panicles, seeds, sterility, and eating quality and stress-related QTLs for drought tolerance and soil stress tolerance. Confirmation of the involvement of the legume lectins in some morphological changes such as growth inhibition and leaf senescence has been reported for OsSIT1 [32]. The gene was identified in QTLs for culm/leaf, dwarfs, panicles/flowers, seeds, lethality and sterility.

### 2.4. Lectin Domain Structure and Phylogenetic Analysis in O. sativa spp.

A detailed analysis of the phylogenetic relationships was performed for the two subspecies of rice, especially, the lectin families which appeared most frequently in our analysis and for which functionally analyzed lectins have been reported in literature. Therefore, the analysis focused on GNA-related lectins, Jacalin-related lectins, and lectins with LysM domains. First, the domain architecture of all lectin sequences within these families was analyzed for *O. sativa* spp. japonica and indica. Second, phylogenetic trees have been constructed for the lectin domain sequences retrieved from *O. sativa* spp. japonica and indica. The phylogenetic trees shown in Figure 4, Figure 5 and Figure 6 were made using all the lectin domain sequences for each of the lectin families under study. The color of the branches reflects the lectin domain combinations. Finally, we also obtained information with respect to duplication events that occurred within each lectin family.

#### 2.4.1. GNA

The GNA family is the largest lectin family in *O. sativa*. As shown in the Appendix A, sequences can consist of the GNA domain only, but more often the GNA domain is linked to other protein domains with known function, such as the PAN (PAN/Apple domain), protein kinase, S-locus receptor kinase, and S-locus glycoprotein domains. These domains can be combined in different configurations resulting in sequences consisting of up to five known protein domains. Some lectin sequences also contain a domain of unknown function (DUF), see Table 1, and Appendix A.

A comparative analysis between the lectin sequences from rice subspecies japonica and indica revealed important differences in the numbers of lectin sequences retrieved from both genomes. The *O. sativa* spp. japonica possesses 134 lectin sequences based on MSU while only 94 sequences with lectin motifs were identified in the genome of *O. sativa* spp. indica [26]. Some lectin domain architectures are unique for one subspecies. The most frequent domain combination GNA/S-locus glycoprotein/PAN/protein kinase is identified in 76 sequences in japonica while only 33 sequences are present in indica. On the other hand, the single GNA sequences as well as the sequences combining GNA/S-locus glycoprotein/PAN are more prevalent in indica compared to japonica. Another combination found only in japonica is a single GNA/S-locus glycoprotein/PAN/protein kinase protein coupled to the Plant organelle RNA recognition domain. According to InterProScan, this domain (IPR021099) is predicted to be localized in the chloroplasts and participates in intron splicing, see Table 1 and Appendix A.

Phylogenetic analysis of the GNA lectin family in *O. sativa* japonica and indica shows some grouping of the sequences based on the domain architecture, see Figure 4. In most sequences, the GNA domain is linked to an S-locus glycoprotein_(1–2)_/PAN/protein kinase domain. All the sequences with GNA/protein kinase (PK) domains cluster together in the phylogenetic tree. A similar observation can be made for the DUF containing proteins from subspecies indica and japonica which cluster closely together as well as GNA/S-locus glycoprotein/PAN/protein kinase/SRK (S-locus receptor kinase). Some clades also cluster GNA domain sequences which are linked to different protein domains, resulting in different domain organizations. For example, the two single GNA domain sequences in japonica do not cluster together. While the one on chromosome 1 (LOC_Os01g72810.1) groups with its homolog in subspecies indica, the other one on chromosome 9 (LOC_Os09g02410.1) groups with GNA/S-locus glycoprotein/PAN/protein kinase and GNA/S-locus glycoprotein/PAN sequences. No clustering was observed for the combinations of GNA/S-locus glycoprotein and GNA/ S-locus glycoprotein/PAN, see Figure 4.

Tandem duplications have played an important role in the expansion of the GNA family in *O. sativa* japonica. In total, 72% of the family members, or 97 genes in 24 duplication blocks, are expanded through this process while the segmental duplications are responsible for 13% (18 genes), see Appendix A. 

#### 2.4.2. Jacalin-Related Lectins

The family of Jacalin-related lectins is the third largest lectin family in *O. sativa*, see Table 1 and Appendix A. The majority of the Jacalin-related lectin sequences consist only of Jacalin domains. In addition, the lectin domain has been duplicated multiple times during evolution giving rise to lectin sequences with two or three tandem repeats of the Jacalin domain. Furthermore, the Jacalin domain has been associated to the NB-ARC (nucleotide binding domain shared by Apaf-1, R Proteins, and CED-4), kinase, peptidase, and dirigent domains, resulting in chimeric sequences consisting of multiple protein domains. The domain architectures with NAM (No apical meristem-associated, C-terminal domain) domain, CTLH (C-terminal LisH motif) domain, and GTPase domain were only found in subspecies indica whereas Jacalin sequences with a peptidase domain have only been retrieved from the genome of subspecies japonica.

As shown in Figure 5, the Jacalin-related sequences in *O. sativa* spp. japonica and indica show some clustering based on their domain organization and are grouped into seven clades in the phylogenetic tree. The first clade comprises most of the multiple Jacalin domain sequences in japonica and the CTLH/Jacalin gene in indica rice. Clades 2, 3, and 6 group lectin sequences with similar domain organizations. Clade 2 combines the genes with a NB-ARC/single Jacalin domain structure, clade 3 includes proteins with single Jacalin domain in their sequence, and clade 6 consists entirely of sequences with a Jacalin domain combined with a dirigent domain. NB-ARC is a motif spread through pro- and eukaryotes and is involved in plant defense as a part of the R (resistance) genes [19]. The combination of a Jacalin domain and a dirigent domain is unique for monocots [27,28], and all proteins with this domain architecture separate in clade 6 of the tree. Clade 4 groups mainly proteins consisting of an NB-ARC domain linked to 3 Jacalin domains as well as proteins composed of a protein kinase domain and 1 to 3 Jacalin domains. Clade 5 and 7 mainly comprise sequences encoding proteins with a single domain structure with few exceptions. 

About 47% of the expansion in the Jacalins in *O. sativa* japonica can be explained by tandem duplications (14 sequences separated in six blocks). The Jacalin-related lectins are not influenced by any segmental duplication events. 

#### 2.4.3. LysM Domain

The LysM domain has undergone several duplication events over time. The majority of the LysM sequences consists of a single LysM domain. Throughout evolution, the LysM domain has been duplicated or has been associated with other protein domains such as the F-box domain or the protein kinase domain.

As shown in Figure 6, the LysM domain sequences in *O. sativa* spp. japonica and indica are grouped into 6 clades in the phylogenetic tree. Similar to the Jacalin-related lectin family, some clustering based on the domain organization can be observed but the domain architecture cannot fully explain the phylogenetic tree organization, see Figure 6. Most of the single LysM domain proteins group together in clades 2, 3, and 6. Clades 1 and 3 consist mainly of LysM/protein kinase domain proteins. LysM/LysM and LysM/LysM/protein kinase sequences group together in clades 2 and 4, forming mixed groups. Intriguingly, the first and second LysM domains are grouped separately in clades 2 and 4, respectively, indication multiple duplication events during evolution. 

Tandem duplications did not have a major impact on the expansion of the LysM lectin family in *O. sativa* japonica, since only 20% of the family members, or only four of the 20 genes, are involved in tandem duplications of the LysM family, see Appendix A. In contrast, segmental duplications had a large impact on the expansion of the family, since 35% of the family members expanded through these segmental duplications. 

## 3. Discussion

Despite the fact that rice is the most investigated crop and multiple resources for research are available, only 1.6% of the rice loci are functionally characterized, in contrast to *Arabidopsis* where 20% of the gene loci have been attributed to a particular function [33]. Although the study of QTLs is a classical method in plant genetics and breeding, there are only a few investigations which link QTLs with the occurrence of specific genes and plant phylogeny. To the best of our knowledge, only one paper investigated the expression profiles and QTL distribution of the Receptor-Like Cytoplasmic Kinase Family in rice [34] but this study only reports the number of the abiotic stress traits in which these proteins were found, without any statistical analysis. Despite the large backlog in rice QTL research and the challenges associated with the characterization of rice QTLs, several lectins have been functionally characterized and their sequences are known to be located in QTL regions. A first example is the mannose-binding lectin from the Jacalin family called SaLT, known as a salt stress and abscisic acid (ABA) responsive protein. A SNP in the lectin sequence results in an amino acid substitution (K24E) that changes the surface charge distribution and most probably influences the protein–protein interactions for this lectin. The haplotype with E24 exhibits a lower Na^+^ concentration in the shoots and this presumably contributes to its higher salt tolerance [13]. A second functionally characterized rice gene is referred to as *Pi-d2* (Os06g0494100) and encodes a GNA-related lectin that confers resistance to rice blast disease. The indica variety Digu, which carries a single substitution 441I, is resistant to *M. grisea* while the varieties with 441M are susceptible to this fungus [14]. The lectin genes for both *SalT* and *Pi-d2* are located in QTLs offering increased salt tolerance or rice blast resistance, respectively, suggesting that the lectin gene might contribute to the stress resistance and/or tolerance attributed to the QTL. 

The current study investigated the presence of lectin sequences in QTL regions in the crop and model species *O. sativa*. Our analysis revealed that lectin sequences are unevenly distributed in QTLs with the highest number of lectin genes on chromosomes 1, 4, and 6. According to Q-TARO, the QTL distribution is unequal over all the 12 rice chromosomes and occurs especially in regions with high gene density. Yonemaru et al. [11] reported that chromosome 1 is the richest in identified QTLs (190), followed by chromosomes 6 and 3 (149 and 146 QTLs, respectively). Six large QTL clusters, each containing more than 15 QTLs, are present on the chromosomes 1, 6, and 3 but some of them also locate to chromosomes 4 and 9 [11].

The putative link between the appearance of lectin sequences in QTLs and different traits for the morphology, physiology, and tolerance/resistance were investigated in detail and statistical analysis was applied. The data provided proof of the overrepresentation of lectin genes in some QTLs associated with important morphological, physiological, and tolerance/resistance traits. Rice is more susceptible to salt stress than other cereals. Since salinity can cause more than 50% yield losses, it is considered as one of the major limitations to grow this crop [5], together with drought which can provoke 53% to 92% yield losses [6]. There are a few examples of lectins involved in salinity tolerance/susceptibility. The best-characterized rice lectins related with salt stress are SaLT, that confers salt tolerance [13], and SIT1 (Os02g0640500) and SIT2 (Os04g0531400), two lectin receptor kinases with a legume lectin domain that has been associated with salt sensitivity. Suppression of SIT1 was found to improve the survival rate under salt stress but impairs plant productivity [32]. Whereas SaLT appeared in our analysis in a QTL for salt stress, none of the *SIT* genes were identified in a QTL region according to the data in Q-TARO. This observation confirms the possibility for the potential presence of additional QTL(s) that have not yet been published or have not been included in the database.

Drought is a trait which influences a broad range of other traits related not only with stress responses but also with grain yield, plant height, germination, fertility, etc. [3]. For example, it is known that the Jacalin-related lectins SaLT and OsJAC1 are responsive to drought stress as well as to cold, salicylic acid, jasmonic acid, and ABA (abscisic acid) stress [27]. The existence of a crosstalk between drought, salt, and cold stress at the level of plant hormones (such as ABA) and transcription factors has been reported in the literature before [35] and is also reflected in the total number of lectins identified simultaneously in more than one type of QTL, see Appendix A. The physical coverage between QTL regions with correlated traits is also observed in other genomes such as the bovine genome, for example, and supports the idea concerning the non-random distribution of genes and clustering of linked genes closer to each other [36]. The presence of lectin sequences in QTLs for both soil stress tolerance and drought tolerance is also expected since a large part of the soil stress includes heavy metal stress (Cd^2+^, Al^3+^, etc.) or nutrient deficiency. Based on transcriptomic data, Oono et al. [37] concluded that a concentration of 50 µM Cd^2+^ causes acute toxicity to rice seedlings and that cadmium stress shares some common signalization with drought stress. This was also in agreement with the observed phenotypic effect on the treated plants which was similar to the effect of drought stress [37].

Even though the number of the lectin genes identified in QTLs for biotic stress is lower than for abiotic stress, some lectins have been reported as key players in the perception and the defense against these pathogens. This is especially true for members of the LysM family as well as Pi-d2 (GNA family). The fact that the *Pi-d2* sequence did not appear in any QTL for biotic stress resistance in our analyses does not exclude a role in pathogen defense since the conferred resistance is reported for a specific rice variety [14]. OsCERK1 and OsCeBiP are well known as chitin receptors working as homo- and heterodimers [38]. Recently, it was reported that OsCERK1 also recognizes bacterial peptidoglycan (PGN) and lipopolysaccharide (LPS) [39] and plays a crucial role in the formation of arbuscular mycorrhiza [40]. Other members of the LysM family referred to as OsLYP4 and OsLYP6 (Os06g0208800) also have a dual function in recognizing chitin and PGN, consequently influencing the response to *Xanthomonas oryzae* (rice bacterial blight) and *Magnaporthe oryzae* (rice blast disease) [41].

OsJAC1 is one of the Jacalin sequences that contains a dirigent domain, a combination typical only for monocots [27,28]. The dirigent domain was reported to be involved in plant defense since it plays an important role in the synthesis of lignins and lignans, and can discriminate between the optical isomers [28]. Furthermore, the dirigent domain appears to be important for the carbohydrate recognition of the Jacalins, at least in maize and sorghum. The deletion of this domain abolishes the ability of the investigated proteins to bind galactose/lactose (β–aggregating factor) in maize or N-acetylgalactosamine in the sorghum homolog [42]. Several publications reported that OsJAC1 is linked to biotic stress responses. After treatment with *M. grisea*, gene expression shows a fast response reaching a maximum at 12 h after treatment. The transcript levels of the Jacalin-related lectin SaLT show an increase at 12 hpi with *M. grisea* and reach a maximum expression level at 24 hpi [27]. Jacalin-related lectins are also associated with enhanced insect tolerance. Overexpression of SaLT in tobacco confers resistance to the aphids *Myzus persicae* Sulzer and *Acyrthosiphon pisum*, and the beet armyworm *Spodoptera exigua* Hübner [29].

The appearance of the same lectins simultaneously in QTLs for abiotic as well as biotic stresses could be indicative for the pleiotropic effect or crosstalk in the signal transduction and regulatory pathways. A dual role for one sequence/protein is also suggested in the literature because of the observed large overlap between molecular responses towards heavy metals as CdSO_4_ (abiotic stress) and incompatible necrotrophs (biotic stress), for example, and the cross-talk and shared molecules in ROS (reactive oxygen species) and MAP (mitogen-activated protein kinase) kinase signaling [43].

In recent years, there has also been evidence for the involvement of particular lectins in plant development, morphology, and physiology and in some cases, possible pleiotropic effects between the dual roles of lectins in defense and development processes have been suggested. The statistically significant appearance of lectin genes in QTLs for seeds is no surprise since many classical lectins have been reported as being abundant seed storage proteins [16]. Few genes have been investigated and functionally characterized for their importance in rice development. Recent research revealed that OsEMSA1 is involved in the development of the embryo sac in rice. The gene is reported to be expressed in most rice organs: root, stem, leaves, panicles, and ovaries and its reduced expression causes abnormalities in the formation of the female gametophyte and pollination and consequently, impairs the development of seeds. RNAi plants possess significantly lower levels of IAA and gibberellins throughout all plant organs in the heading stage in comparison to wild-type plants [44].

Studies by Zou et al. and Cheng et al. [30,31] showed that lectin receptor-like kinases belonging to the GNA family are also associated with rice morphology and physiology. OsLSK1 is a lectin receptor-like kinase which possesses an extracellular GNA lectin domain, a PAN domain, an S-locus domain, and an intracellular protein kinase domain, and has six homologs in the rice genome which can form hetero- or homodimers. Based on its expression analysis, it is hypothesized that this gene is important in node development. Transcript levels are upregulated after treatment with gibberellins and brassinosteroids while treatments with polyethylene glycol and salt cause fast downregulation of gene expression. The truncated form lacking the kinase domain leads to a significant increase in the number of primary branches in the panicles, in the number of grains per branch, and in plant height. The appearance of all OsLSK1 sequences in QTLs for seeds, drought tolerance, and salinity tolerance is in agreement with these data. Knockdown of the transcript levels of another receptor-like kinase, OslecRK, leads to reduced seed germination and impaired immunity towards a broad spectrum of pathogens, such as *M. grisea*, *Xanthomonas oryzae*, and the insect brown planthopper (*Nilaparvata lugens*), to downregulation of some defense-related genes, and to a reduction in α-amylase enzymes during germination. These effects can be explained partially from the identified interaction between the kinase domain and actin-depolymerizing factors but most probably the gene acts through other interactors as well [31]. 

The already mentioned, OsSIT1, which influences salt tolerance/susceptibility through the ethylene signaling pathway, is linked to some morphological changes such as growth inhibition and leaf senescence when it is overexpressed. Delayed maturation and impaired productivity were observed in knockdown lines not only in rice but also in *Arabidopsis* [32], which is in agreement with the present study and the presence of the lectin gene in QTL regions associated with the traits culm/leaf, dwarfs, panicles/flowers, seeds, lethality, and sterility.

Some Jacalin-related lectins are associated with developmental functions in addition to their involvement in stress responses. Overexpression of OsJAC1 reduces the cell length which in turn results in shorter internodes, shorter coleoptiles, and a dwarf phenotype [45]. Additional confirmation of the involvement of the Jacalins in plant development comes from the AtJAC1 gene from Arabidopsis, which is one of the negative regulators of flowering time, and from VER2 in wheat, which is a positive regulator of flowering after vernalization [46].

In addition to the in-depth analysis of lectin sequences in QTL regions, a detailed study was performed on the presence and the evolution of lectin sequences in two subspecies of *O. sativa.* Comparative analyses revealed some unique domain combinations as well as quantitative differences between the subspecies indica and japonica, mainly for the lectin motifs in the largest lectin families (GNA, Legume lectin, Jacalin, and LysM). Most lectin motifs are linked to other protein domains with known function. GNA and Legume lectins appeared predominantly in combination with protein kinase domains. Taking into account the localization of many of these sequences in the cell wall, these proteins probably play an important role in the perception of extracellular signals and are the first players in signaling cascades as part of plant immune response. Such functions have already been reported for Pi-d2, OsSIT1, and OsLSK1 [14,31,32]. Although the focus of this study was mainly on the differences between the subspecies japonica and indica, it is also worth mentioning that Hevein and EUL sequences are highly conserved between both subspecies, suggesting an ancient origin and important physiological role of these lectins.

The presence of a large collection of lectin motifs distributed over QTL regions raises questions on the carbohydrate-binding activity and how protein–carbohydrate interactions contribute to the biological significance of the lectin domains. Different lectin domains can recognize very diverse mono- and oligosaccharides. For instance, the hevein and the LysM domain both recognize and bind chitin and chito-oligosaccharides. Lectins from the GNA family are mainly mannose-specific but also interact with mannosylated *N*-glycan structures. Within the Jacalin family, two subclasses of lectins with specificity towards mannose and galactose have been reported. The legume lectin motif can interact with very diverse carbohydrate structures since the shape of the carbohydrate binding site will change due to different positioning of a flexible loop. These data illustrate that it is impossible to predict the carbohydrate binding properties based on sequence information. Ultimately the carbohydrate binding specificity for each lectin should be analyzed experimentally. In recent years, it has become clear that most lectin sequences are composed of multiple protein domains and can be considered chimeric lectins in which a lectin domain is linked to one or more protein domains, indicating that the biological activity of these proteins will depend on the activity and possible interplay of multiple protein domains. Future experiments will need to investigate the physiological importance of the lectin domains in rice.

In conclusion, at present, little is known with respect to the physiological role of lectin sequences. Our statistical analyses showed that lectin sequences are overrepresented in several QTLs, suggesting that the lectins sequences could play a role in the trait(s) attributed to the QTLs. Since one gene can have pleiotropic effects, more in-depth studies are needed to provide more insights regarding the physiological role of the lectins, possible pleiotropic effects, and how to integrate and use these data in practice. Furthermore, the possibility that lectins are involved in or are important for many more traits cannot be ruled out since this research was based on a database that is probably incomplete, simply because not all QTLs are known at present. Our data contribute to the knowledge of lectin functionality and the importance of lectins for plant growth and development. By combining QTL analyses and genome analyses, we could link some genetic information with physiological and biochemical studies, aiming for a better functional understanding of plant lectins.

## 4. Materials and Methods

### 4.1. Identification of Putative Lectin Genes in the Genome of O. sativa spp. Indica and Japonica

The identification of putative lectin genes in the rice genome was previously described by De Schutter et al. [26]. In short, protein sequences of reference members for the different lectin families were used in BLAST searches against the *O. sativa* spp. japonica genome. The lectin sequences were screened for the presence of conserved protein domains to confirm the lectin domain and determine the domain organization. Only sequences with the conserved lectin domains were retained in the analysis. The protein sequences retrieved from the *O. sativa* spp. japonica were used to identify orthologs in the genome of *O. sativa* spp. indica using BLAST searches [26]. The MSU IDs for all identified lectin sequences were converted to RAP-DB IDs using the RAP-DB ID convertor tool (https://rapdb.dna.affrc.go.jp/tools/converter).

### 4.2. Analysis of Segmental and Tandem Duplications in O. sativa spp. Japonica

Gene expansion through segmental and tandem duplications was analyzed for the japonica sequences [26]. Tandem duplications were defined as two genes belonging to the same family and located on the same chromosome with a maximum of 10 intervening genes. Segmental duplications were identified using the Plant Genome Duplication Database (PGDD, http://chibba.agtec.uga.edu/duplication) [47]. The dataset was filtered to discard duplicated genes with a Ks (synonymous substitution) value higher than 1.0. The remaining data were searched for lectin genes.

### 4.3. Mapping of the Lectin Genes on the Rice Chromosomes and in QTL Regions

The lectin genes from *O. sativa* japonica were mapped on the chromosomes using the MapChart software (v2.32, https://www.wur.nl/en/show/Mapchart.htm) [48]. The information about the QTL positions was downloaded from Q-TARO database (http://qtaro.abr.affrc.go.jp/) and mapped with the same tool using RAP-DB gene annotations since Q-TARO is based on this database. Overlaps of genomic loci between QTL and lectin genes were determined for every combination of lectin families and QTL group using the intersect function of the bedtools utilities (v2.27.1, https://bedtools.readthedocs.io/en/latest/) [49]. The information on QTLs linked to nematode *Meloidogyne graminicola* resistance was retrieved from Dimkpa et al. [15] since this information is not included in Q-TARO.

### 4.4. Statistical Analysis

Every trait was analyzed for the presence of redundant QTL. All overlapping QTL were merged into a non-redundant single QTL and all redundant QTL were excluded from further analysis [36]. The identification of the expressed genes which are located in the non-redundant QTL was performed using the RAP-DB batch retrieval tool (https://rapdb.dna.affrc.go.jp/tools/dump). An additional step involving the filtering of the redundant transcripts was applied to provide the total number of the unique gene IDs. During our analysis, QTLs which do not contain any lectin or another type of gene were identified and excluded from all subsequent computations.

The significant overrepresentation of lectin genes was determined with the use of a Wald test in which the null hypothesis states an absence of lectin overrepresentation, i.e., lectins are not preferentially located on the QTL. The *z*-score was calculated as the proportion of lectin genes in the genome (*p_g_*) subtracted from the QTL proportion of lectin genes for a specific trait (*p_trait_*), divided by the standard error on *p_trait_*. The latter can be computed from the hypergeometric distribution due to the fact that lectin genes occur once in the genome. Hence,
$$z=\frac{p_{trait}-p_{g}}{SE\left(p_{trait}\right)}$$ where
$$var\left(p_{trait}\right)=\frac{var\left(x\right)}{n^{2}}$$ and
$$var\left(x\right)=\left(\frac{nsr}{N^{2}}\right)\times \left(\frac{N-n}{N-1}\right)$$

*n*—number of the genes in QTL; *s*—number of the lectins in the genome; *r*—number of the non-lectin genes in the genome; *N*—total sample (all genes in the genome); *x*—number of lectin genes (in the trait).

We considered *N* as being all protein coding sequences in the genome of *O. sativa* spp. japonica according IRGSP-1.0 published in 2016 (https://www.ncbi.nlm.nih.gov/genome/annotation_euk/Oryza_sativa_Japonica_Group/101/#FeatureCountsStats) with the total number of genes in QTL (*N*) equal to 27,912 and the total number of the lectin genes in the genome (*s*) equal to 291 (according to RAP-DB).

A Bonferroni correction was applied that considered multiple testing. A p-value was considered as significant if *p* < 0.001, where the significance level, α, for a single two-sided test was defined as 0.025 and was divided by the number of performed tests, i.e., 25.

### 4.5. Phylogenetic Analysis

Maximum-likelihood phylogenetic trees were generated using the lectin domain sequences identified for different lectin families according to the MSU annotation project, as described by [20]. For sequences consisting of multiple lectin domains, each lectin domain was treated as a separate entry in the analysis. The domain sequences were aligned in MEGA7 (https://www.megasoftware.net/) with MUSCLE using default settings [50]. Consecutive trimming of the aligned sequences was performed using the automated1 option of trimAl v. 3 (http://trimal.cgenomics.org/) [51]. Phylogenetic trees were built with RAxML v. 8.2.4 based on the trimmed alignments [52]. RAxML used the GTRGAMMA model with an automated determination of the best amino acid substitution model. Bootstrap iterations were generated to assess the robustness of the tree and the number of bootstraps was decided automatically by RAxML up to a maximum of 1000. Phylogenetic trees were visualized and annotated with FigTree v. 1.4.2 software (http://tree.bio.ed.ac.uk/software/figtree/).

## Figures and Tables

**Figure 1 ijms-20-00437-f001:**
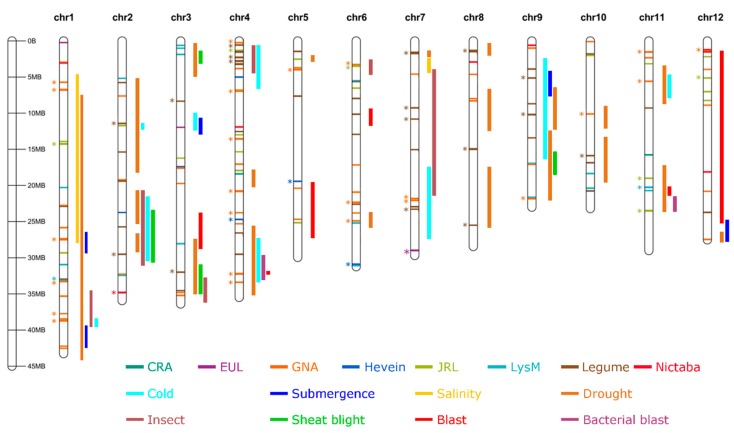
Localization of lectin genes and quantitative trait loci (QTLs) for resistance/tolerance on rice chromosomes of *O. sativa* spp. japonica. Lectins are indicated by horizontal bars inside the chromosomes, and are shown in distinct colors. Tandem duplicated blocks are indicated by asterisks. QTLs are indicated by vertical bars next to the chromosomes, represented in different colors.

**Figure 2 ijms-20-00437-f002:**
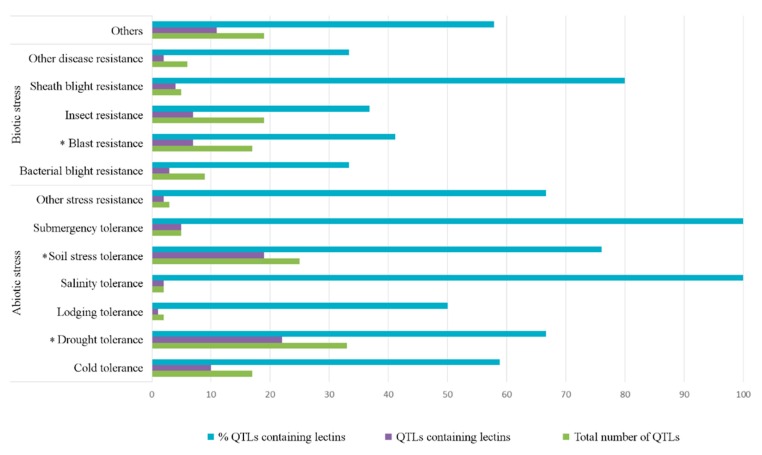
Total number of QTLs for resistance/tolerance, number of QTLs for resistance/tolerance containing lectin sequences, and percentage of QTLs for resistance/tolerance containing lectin sequences. The asterisks refer to statistically significant overrepresentation (*p* < 0.001) of lectin sequences in QTLs for the trait(s).

**Figure 3 ijms-20-00437-f003:**
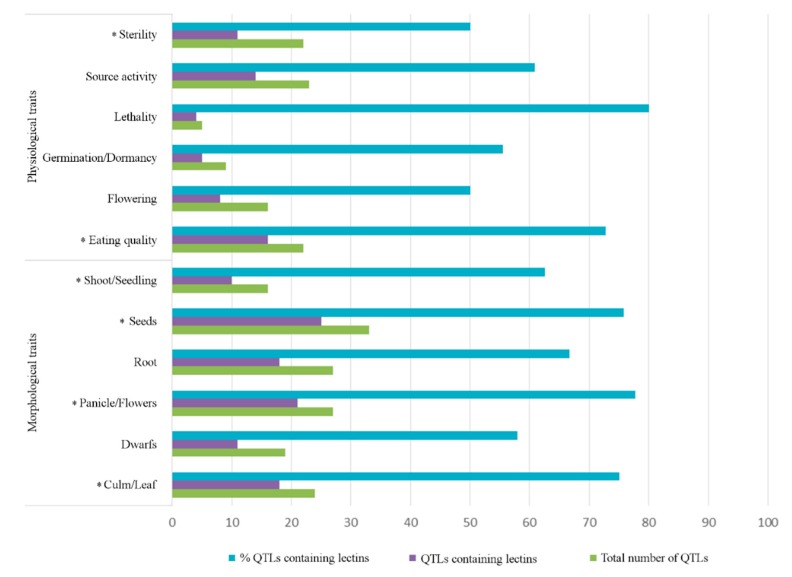
The total number of QTLs for morphological and physiological traits, number of QTLs for morphological and physiological traits containing lectin sequences, and percentage of QTLs for morphological and physiological traits containing lectin sequences. The asterisks refer to a statistically significant overrepresentation (*p* < 0.001) of lectin sequences in QTLs for the trait(s).

**Figure 4 ijms-20-00437-f004:**
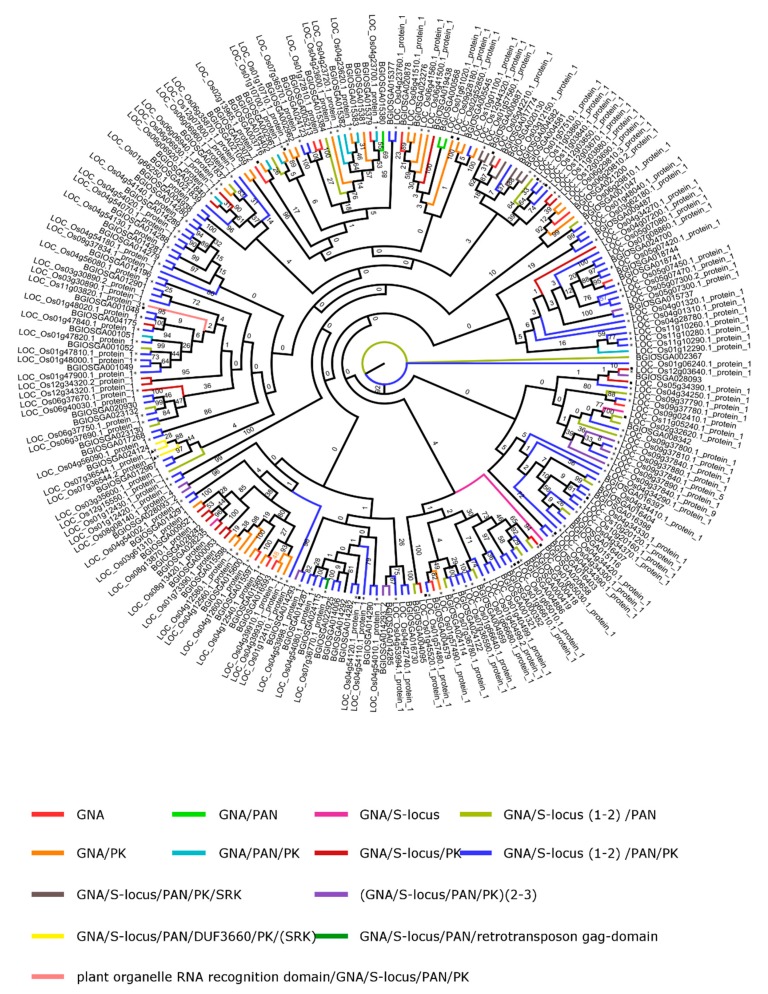
Phylogenetic tree of the GNA lectin family. Asterisks designate a tandem duplication event. Triangles refer to a single lectin domain in a multiple domain structure. Squares designate a segmental duplication event. The black numbers indicate the bootstrap values while the grey ones indicate the number of the clades. The numbers separated with underscore from the ID of the genes are referring to the number of the lectin domain in sequences that contain multiple lectin domains.

**Figure 5 ijms-20-00437-f005:**
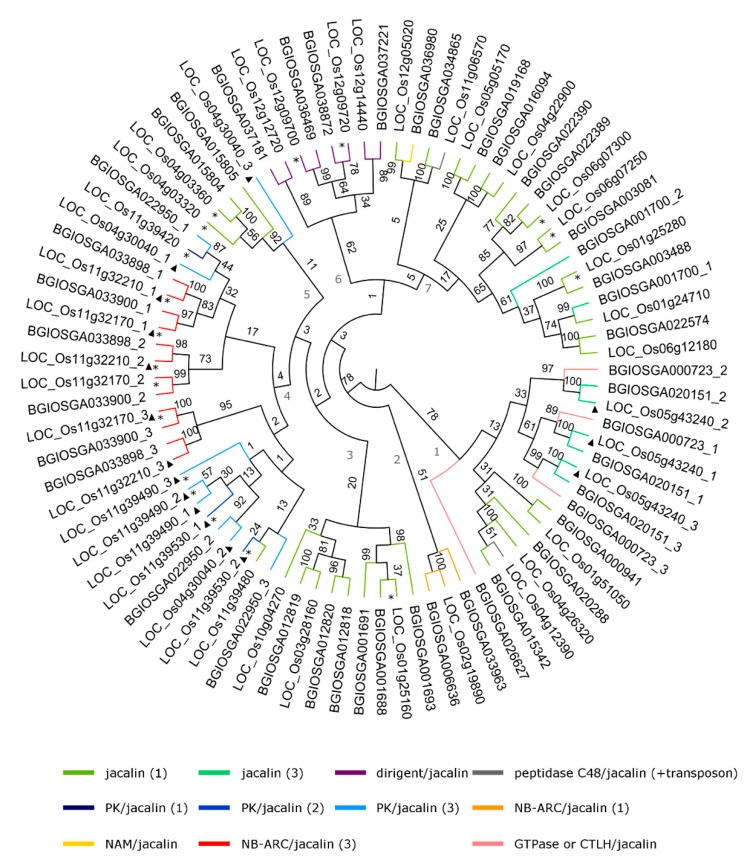
Phylogenetic tree of the Jacalin lectin family. Asterisks refer to tandem duplication events whereas triangles represent a single lectin domain in a multiple domain structure. The black numbers indicate the bootstrap values while the grey ones indicate the number of the clades. The numbers separated by an underscore from the ID of the genes are referring to the number of the lectin domain in multiple domain sequences.

**Figure 6 ijms-20-00437-f006:**
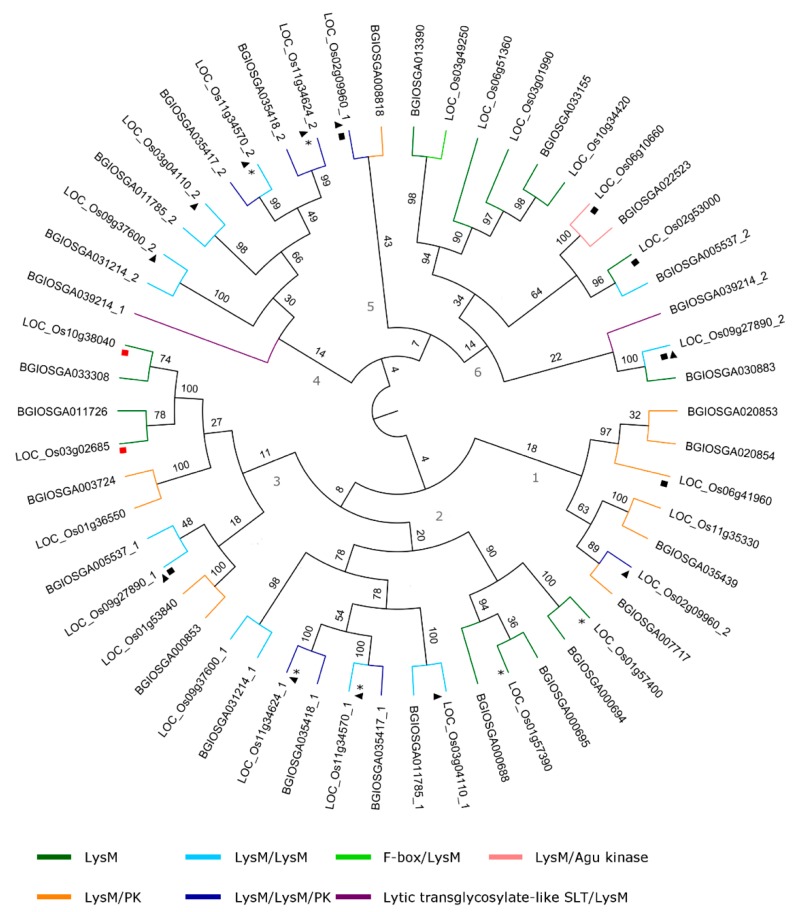
Phylogenetic tree of LysM lectin family. Asterisks refer to a tandem duplication event. Triangles represent a single lectin domain in a multiple domain structure. Black squares show segmental duplication events whereas red squares show segmental duplication events where only one of the duplicated segments contains a lectin gene. The black numbers indicate the bootstrap values while the grey ones indicate the number of the clades. The numbers separated with underscore from the ID of the genes are indicating the number of the lectin domain in multiple domain sequences.

**Table 1 ijms-20-00437-t001:** Schematic representation of the shared domain architectures in *O. sativa* spp. japonica and indica. The numbering and domain organization for japonica is based on Rice Annotation Project Database (RAP-DB). * domain organization of the most representative combinations in the *Galanthus nivalis* agglutinin (GNA) family, see Appendix A. ** includes also multiple repeat domain proteins. *** DUF (domain of unknown function) 3660 in spp. japonica, DUF 3404 in spp. indica.

Lectin and Domain Architecture	Schematic Representation	Spp. Japonica	Spp. Indica
CRA	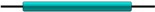	2	3
EUL	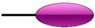	2	2
(EUL)_2_	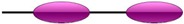	3	3
Hevein	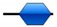	1	1
(Hevein)_4_		1	1
Hevein/GH19	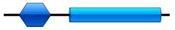	8	8
Nictaba	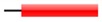	4	5
F-box/Nictaba	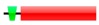	13	14
Fbox/(Nictaba)_2_	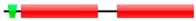	1	2
Protein kinase/Nictaba	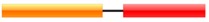	1	1
Jacalin	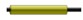	16	21
(Jacalin)_2_	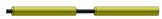	1	1
(Jacalin)_3_	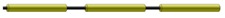	1	1
Protein kinase/Jacalin	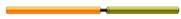	1	0
Protein kinase/(Jacalin)_2_	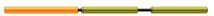	1	0
Protein kinase/(Jacalin)_3_	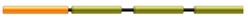	2	1
Dirigent/Jacalin	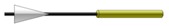	4	4
NB-ARC/Jacalin	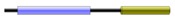	2	2
NB-ARC/(Jacalin)_3_	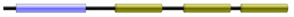	2	2
LysM	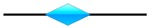	8	8
(LysM)_2_	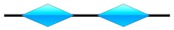	4	3
LysM/Protein kinase	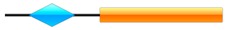	4	7
(LysM)_2_/Protein kinase	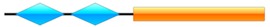	1	1
Aspartate/glutamate/uridylate kinase/LysM	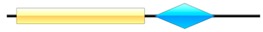	2	2
Legume lectin	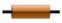	14	12
(Legume lectin)_2_	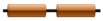	1	0
Legume lectin/Protein kinase	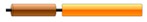	61	73
(Legume lectin)_2_/Protein kinase	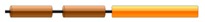	1	2
GNA	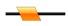	2	11
GNA/(S-locus glycoprotein)_1–2_/(PAN)/(protein kinase)_1–2_/(SRK) *	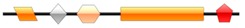	119	82 **
GNA/S-locus glycoprotein/PAN/DUF ***/protein kinase/(SRK)	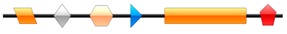	2	1
Legend
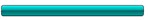	CRA (PF00704)		GH19 (PF00182)
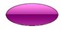	EUL (PF14200)	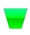	F-box (PF00646)
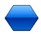	Hevein (PF00187)		Protein kinase domains (PF00069, PF50011, PF07714)
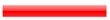	Nictaba (PF14299)	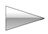	Dirigent (PF03018)
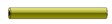	Jacalins (PF01419)	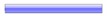	NB-ARC (PF00931)
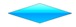	LysM (PF01476)	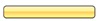	Aspartate/glutamate/uridylate kinase (PF00696)
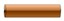	Legume lectin (PF00139)	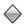	S-locus glycoprotein (PF00954)
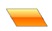	GNA (PF01453)	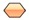	PAN domains (PF00024, PF08276, PF14295)
		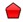	SRK (PF11883)
		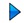	DUF domains (PF12398, PF11884)

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
