# Peer review of "Lectin Sequence Distribution in QTLs from Rice (Oryza sativa) Suggest a Role in Morphological Traits and Stress Responses"

_ijms, 2019, doi:10.3390/ijms20020437_

Round 1
Reviewer 1 Report
The authors describe the lecin information of rice (Oryza sativa) by analyzing quantitative trait loci (QTL) towards crop-related protein availability because some lectins are stress inducible during cultivation. Distribution of lectin sequences in the stress tolerance or resistance is explained with its morphology and physiology based on the chromosomes. Some results are statistically analyzed, with poor accuracy.
Moreover, the structure and protein similarity of the lectins are compared in the O. sativa indica and japonica classes. Interesting results are deduced from the lectin sequences
in resistance, tolerance, quality and sterility, allowing the breedinf papameters in future crop science.
The study is valuable for ducumentation due to its validity in rice crop breeding towards the environmental tolerance or pathogenic resistance. However, the study and description are suffered from the non-well designed presentation of the results.
Statistics in Figures: The data are expressed as the % level without the sufficient number od analysis affording the accurate statistical significance. All the data should be reanalyzed with the ststistics with at least 3 times analysis even in the restricted approach.
The lectin informations are limited in the definition of carbohydrate-binding capacity. Although the keywords of lectins are understood, the additional information of each lectin candidate should be explained for the carbohydrate-binding property. FOr example, the chitin-binding liknage should be described in ech Table.
Generally, this is an interesitng approach in ths plant lectin diversity and evolutionary linakges withe the same species during the different adaptation to environment.
After approprate revision, this can be accepted.
Author Response
We have made a revised text taken into account the comments and suggestions of the reviewer.
1. The results section has been changed considerably. Results have been separated from the discussion. Discussion was combined with conclusions.
2. Information on the possible carbohydrate-binding activity is lacking for most lectins from rice. We have added a new paragraph dealing with carbohydrate-binding properties and biological activity of the rice lectins in the introduction and in the discussion of the revised manuscript.
3. With respect to statistics: The dataset which we have did not allow the application of classical statistical methods with 3 replicates because we did not perform a laboratory experiment. Since the data for the QTL distribution are obtained from a database (Q-TARO; http://qtaro.abr.affrc.go.jp/ ), which represents manually curated results, we have no access to the initial calculations and the p-values for the identification of the QTLs. Because we observed that some QTLs were redundant or/and overlapping, all overlapping QTLs were merged in non-redundant single QTLs and all redundant QTLs were excluded from further analysis (Salih and Adelson, 2009). A graphical representation of the total number of the lectin genes towards the total number QTLs can be done but: 1) can be confusing for people who refer themselves to the total number of QTLs published in Q-TARO; 2) does not give the idea how many of the QTLs per trait comprise at least 1 lectin gene and can give wrong idea that every QTL which is considered contains lectin genes, which is not the case.
As indicated in the acknowledgements section of the manuscript, the statistical analysis shown in the manuscript was performed after consulting with Ghent University FIRE (Fostering Innovative Research based on Evidence) and according to their advice we applied a Wald test for comparing proportions (pg vs ptrait). This test belongs to a group of parametrical tests allowing rejection of the null hypothesis, in our case the null hypothesis states an absence of lectin overrepresentation, i.e. lectins are not preferentially located on the QTLs. Stated differently, the lectin QTL-proportion is not significantly different from the genome-proportion. All tests from this group base their inference about the parameters on the sample data and size (Agresti, 2003). Because the Wald test is not correct for small sample size (which for the Wald test is considered as<30), all lectin sequences were investigated together rather than for each individual family separately. Only the two largest lectin families (GNA and Legume lectins) comprise a number of lectin sequences that is much higher than 30. In addition, proportions close to 0, as is the case in this study, need more care too. For that reason, the Agresti-Caffo correction was applied (Agresti & Caffo, 2000).
z (z-score) is approximately normally distributed in a large sample size (Agresti, 2003). In our case, we applied two-tailed (two-sided) test where α taken as 0.025 for a two-sided test, was subjected to a Bonferroni correction for multiple testing. Since we performed 25 independent tests (25 different QTL categories), α is divided by the number of the performed tests, i.e. 25. In this way we obtained a p value = 0.001 and the null hypothesis was rejected at p < 0.001 which coincides with the p value used in most of the tests.
The values from the intermediate steps of our analysis are represented in Suppl. Table S7 but if
the reviewer considers more info is needed to allow repetition of the same analysis, we
would be happy to receive advice how the information can be completed.
References:
Agresti, A. (2003). Categorical data analysis (Vol. 482). John Wiley & Sons.
Agresti, A. & Caffo, B. (2000). Simple and effective confidence intervals for proportions and difference of proportions result from adding two successes and two failures. The American Statistician, 54, 280–288.
Salih, H., & Adelson, D. L. (2009). QTL global meta-analysis: are trait determining genes clustered? BMC genomics, 10(1), 184.
Reviewer 2 Report
It is an excellent study that revealed the presence of numerous lectin domain genes on the chromosome by using rice.
In particular, it is valuable on the description that the expression of lectin gene consisting of various domain structures is different based on the difference between Indica and Japonica, which are two subspecies of rice.
It will be able to consider the relationship between lectin diversity and evolution.
Furthermore, authors also indicated the relationship between the lectin gene and the resistance to diseases, relating with the food problem considering sasteinable development.
It will provide a new viewpoint from lectinology to plant breeding and genetics.
On the other hand, the ”Results and Discussion" part has an impression that the results and considerations are written in parallel, making it difficult to read the comparisons throughout the results and the consideration. Reviewer thinks improving it by the authors will lead to increase the value of this paper.
Improvement points:
# Since Abstract part says that ”lectin has diverse functions” (p1 L 18), writing about breaf introduction of innate immunity and others as functions of plant lectins can provide views of importance on plant lectins to the readers at Introduction part (p2 Line 82).
Since the authors are the representative research group on this subjects, readers want to know about it easily in this paper.
# In Table 1, which occupies an important part in this paper, the lines of EUL and (EUL)2, Hevein and (Hevein)4 as same as other lectins should separate to avoide confusion.
# The line under "Schematic representation" disappears by the picture of "CRA".
# Add "subspecies" on the column "Japonica" and "Indica" as many readers even if do not major on plant breeding and genetics can understand the meaning of the Table.
# Using the same colors on each lectin in Fig 1 and Table 1 helps the reader understand.
# By showing the notes in Table 1, which lectin each symbol represents, together with the classification of pfam, will give the reader more empathy.
# ”Results and Discussion" part are mixed both in the result and discussion, which hinders to understand the significance of the study to common readers. And the Conclusion (about 500 words) part more clearly describes the value on this research.
Considering about these, the reviewer thinks that it will be available to divide "Results" part, from "Results and Discussion" and combine a part of "Conclusion" to Discussion to enrich "Discussion" part to inform readers about the significance of this study. New "Conclusion" may be readable if it is much volume as same as "Abstract".
# The reviewer hope to add the authors' thought or your speculation about the target glycans and the organisms to which these lectins bind in discussion part to conclude the function of these diverse lectins.
Author Response
We have made a revised text taken into account the comments and suggestions of the reviewer.
1. The results section has been changed considerably. Results have been separated from the discussion. Discussion was combined with conclusions.
2. Figure 1 and Table 1 have been adapted. The lectin domains are now shown in the same colors in the figure and the table. A legend has been added to Table 1 to explain the symbols used and indicate the Pfam identifiers for each protein domain, as requested by the reviewer.
3. Information on the possible carbohydrate-binding activity is lacking for most lectins from rice. We have added a new paragraph dealing with carbohydrate-binding properties and biological activity of the rice lectins in the introduction and in the discussion of the revised manuscript.
Round 2
Reviewer 1 Report
Figures 2 and 3 should be updated, as I raised the same issue in the first line of review. The present figures are not of high calibre
Author Response
New figures (Figures 2 and 3) have been introduced in the manuscript.